# Test-Time Collective Prediction

**Celestine Mendler-Dünner**[*]
MPI for Intelligent Systems, Tübingen
`cmendler@tuebingen.mpg.de`

**Wenshuo Guo**
University of California, Berkeley
`wguo@cs.berkeley.edu`

**Stephen Bates**
University of California, Berkeley
`stephenbates@cs.berkeley.edu`

**Michael I. Jordan**
University of California, Berkeley
`jordan@cs.berkeley.edu`

## Abstract

An increasingly common setting in machine learning involves multiple parties, each with their own data, who want to jointly make predictions on future test points. Agents wish to benefit from the collective expertise of the full set of agents to make better predictions than they would individually, but may not be willing to release labeled data or model parameters. In this work, we explore a decentralized mechanism to make collective predictions at test time, that is inspired by the literature in social science on human consensus-making. Building on a query model to facilitate information exchange among agents, our approach leverages each agent's pre-trained model without relying on external validation, model retraining, or data pooling. A theoretical analysis shows that our approach recovers inverse mean-squared-error (MSE) weighting in the large-sample limit which is known to be the optimal way to combine independent, unbiased estimators. Empirically, we demonstrate that our scheme effectively combines models with differing quality across the input space: the proposed consensus prediction achieves significant gains over classical model averaging, and even outperforms weighted averaging schemes that have access to additional validation data. Finally, we propose a decentralized Jackknife procedure as a tool to evaluate the sensitivity of the collective predictions with respect to a single agent's opinion.

## 1 Introduction

Large-scale datasets are often collected from diverse sources, by multiple parties, and stored across different machines. In many scenarios centralized pooling of data is not possible due to privacy concerns, data ownership, or storage constraints. The challenge of doing machine learning in such distributed and decentralized settings has motivated research in areas of federated learning [Konečný et al., 2015, McMahan et al., 2017], distributed learning [Dean et al., 2012, Gupta and Raskar, 2018], as well as hardware and software design [Zaharia et al., 2016, Moritz et al., 2018].

While the predominant paradigm in distributed machine learning is *collaborative learning* of one centralized model, this level of coordination across machines at the training stage is sometimes not feasible. In this work, we instead aim for *collective prediction* at test time without posing any specific requirement at the training stage. Combining the predictions of pre-trained machine learning models has been considered in statistics and machine learning in the context of ensemble methods, including bagging [Breiman, 1996a] and stacking [Wolpert, 1992]. In this work, we explore a new perspective on this aggregation problem and investigate whether insights from the social sciences on how humans reach a consensus can help us design more effective aggregation schemes that fully take advantage of

---

[*]most of the work conducted while at UC Berkeley.

35th Conference on Neural Information Processing Systems (NeurIPS 2021).

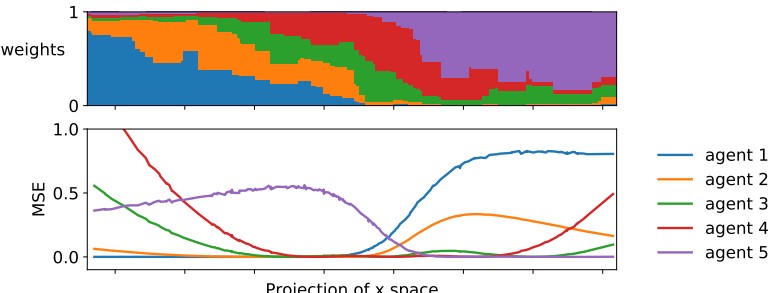

Figure 1: *(Test-time aggregation of heterogeneous models).* The proposed method upweights more accurate models (lower MSE) adaptively in different regions of the input space. Constructed from a regression task described in Section 4.1.

each model's individual strength. At a high level, when a panel of human experts come together to make collective decisions, they exchange information and share expertise through a discourse to then weigh their subjective beliefs accordingly. Experts have a conforming influence on each other which gives rise to a dynamic process of consensus finding that is decentralized and does not rely on any external judgement. We map this paradigm to a machine learning setting where experts correspond to pre-trained machine learning models, and show that it leads to an appealing mechanism for test-time collective prediction.

## 1.1 Our work

We build on a model for human consensus finding that defines an iterative process of opinion pooling based on *mutual trust scores* among agents [DeGroot, 1974]. Informally speaking, a mutual trust score reflects an agent's willingness to adapt another agent's opinion. Thus, each individual's impact on the final judgment depends on how much it is trusted by the other agents.

Mapping the DeGroot model to our learning scenario, we develop a scheme where each agent uses its own local training data to assess the predictive capabilities of other agents' models and to determine how much they should be trusted. This assessment of mutual trust is designed to be adaptive to the prediction task at hand, by using only a subset of the local data in the area around the current test point. Then, building on these trust scores, a final judgment is reached through an iterative procedure, where agents repeatedly share their updated predictions.

This aggregation scheme is conceptually appealing as it mimics human consensus finding, and it exhibits several practical advantages over existing methods. First, it is decentralized and does not require agents to release their labeled data or model parameters. This can be beneficial in terms of communication costs and attractive from a privacy perspective, as it allows agents to control access to their data and model at any time. Second, it does not require hold-out validation data to construct or train the aggregation function. Thus, there is no need to collect additional labeled data and coordinate shared access to a common database. Third, it does not require any synchronization across agents during the training stage, so agents are free to choose their preferred model architecture, and train their models independently.

A crucial algorithmic feature of our procedure is its *adaptivity* to the test point at hand, which allows it to deal with inherent variations in the predictive quality of models in different regions of the input space. Figure 1 illustrates how our mechanism adaptively upweights models with lower error depending on the test point's location in the input space. In fact, we prove theoretically that our aggregation procedure can recover inverse-MSE weighting in the large-sample limit, which is known to be optimal for variance reduction of unbiased estimators [Bates and Granger, 1969].

To assess our procedure's sensitivity to the predictions of individual agents we propose a decentralized Jackknife algorithm to compute error bars for the consensus predictions with respect to the collection of observed agents. These error bars offer an attractive target of inference since they are agnostic to how individual models have been trained, can be evaluated without additional model evaluations, and allow one to diagnose the vulnerability of the algorithm to malicious agents.

On the empirical side, we demonstrate the efficacy of our mechanism through extensive numerical experiments across different learning scenarios. In particular, we illustrate the mechanism's advantages over model averaging as well as model selection, and demonstrate that it consistently outperforms alternative non-uniform combination schemes that have access to additional validation data across a wide variety of models and datasets.

## 1.2 Related work

The most common approach for combining expert judgment is to aggregate and average individual expert opinions [Hammitt and Zhang, 2013]. In machine learning this statistical approach of equally-weighted averaging is also known as the "mean-rule" and underlies the popular *bagging* technique [Breiman, 1996a, Buja and Stuetzle, 2006, Efron, 2014]. It corresponds to an optimal aggregation technique for dealing with uncertainty in settings where individual learners are homogeneous and the local datasets are drawn independently from the same underlying distribution. While principled, and powerful due to its simplicity, an unweighted approach cannot account for heterogeneity in the quality, or the expertise, of the base learners.

To address this issue, performance-weighted averaging schemes have been proposed. For determining an appropriate set of weights, these approaches typically rely on calibrating variables [Aspinall, 2010] to detect differences in predictive quality among agents. In machine learning these calibrating variables correspond to labeled cross-validation samples, while in areas such as expert forecasting and risk analysis these variables take the form of a set of agreed-upon seed questions [Cooke, 1991]. These weights can be updated over time using statistical information about the relative predictive performance [Rogova, 2008] or the posterior model probability [Wasserman et al., 2000]. More recently there have been several applications to classification that focus on dynamic selection techniques [Cruz et al., 2018b], where the goal is to select the locally best classifier from a pool of classifiers on the fly for every test point. At their core, these approaches all rely on constructing a good proxy for local accuracy in a region of the feature space around the test point and to use this proxy to weight or rank the candidate models. Various versions of distance and accuracy metrics have been proposed in this context [eg. Woods et al., 1997, Didaci et al., 2005, Cruz et al., 2018a]. However, all these methods rely crucially on shared access to labeled validation data to determine the weights. Similarly, aggregation rules such as *stacking* [Breiman, 1996b, Coscrato et al., 2020] or *meta-models* [Albardan et al., 2020], both of which have proved their value empirically, require a central trusted agency with additional data to train an aggregation function on top of the predictor outputs. A related model-based aggregation method, the mixture-of-experts model [Jacobs et al., 1991], requires joint training of all models as well as data pooling.

To the best of our knowledge there is no approach in the literature that can perform adaptive performance weighting without relying on a central agent to validate the individual models. Notably, our approach differs from general frameworks such as model parameter mixing [Zhang et al., 2013], model fusion [Leontev et al., 2018, Singh and Jaggi, 2020] in that agents are not required to release their models.

Outside machine learning, the question of how to combine opinions from different experts to arrive at a better overall outcome has been studied for decades in risk analysis [Hanea et al., 2018, Clemen and Winkler, 1999] and management science [Morris, 1977]. Simplifying the complex mechanism of opinion dynamics, the celebrated DeGroot model [DeGroot, 1974] offers an appealing abstraction to reason mathematically about human consensus finding. It has been popular in the analysis of convergent beliefs in social networks and it can be regarded as a formalization of the Delphi technique, which has been used to forecast group judgments related to the economy, education, healthcare, and public policy [Dalkey, 1972]. Extensions of the DeGroot model have been proposed to incorporate more sophisticated socio-psychological features that may be involved when individuals interact, such as homophily [Hegselmann and Krause, 2002], competitive interactions [Altafini, 2013] and stubbornness [Friedkin and Bullo, 2017].

## 2 Preliminaries

We are given $K$ independently trained models and aim to aggregate their predictions at test time. Specifically, we assume there are $K$ agents, each of which holds a local training dataset, $\mathcal{D}_k$, of size

$n_k$, and a model $f_k : \mathcal{X} \to \mathcal{Y}$. We do not make any prior assumption about the quality or structure of the individual models, other than that they aim to solve the same prediction task.

At test time, let $x' \in \mathcal{X}$ denote the data point that we would like to predict. Each agent $k \in [K]$ has her own prediction for the test point corresponding to $f_k(x')$. Our goal is to design a mechanism $\mathcal{M}$ that combines these predictions into an accurate single prediction, $p^*(x') = \mathcal{M}(f_1(x'), ..., f_K(x'))$. In the following we focus on regression and use the squared loss as a measure of predictive quality.

The requirements we pose on the aggregation mechanism $\mathcal{M}$ are: (i) agents should not be required to share their labeled data or model parameters with other agents, or any external party; (ii) the procedure should not assume access to additional training data for an independent performance assessment or training of a meta-model.

### 2.1 The DeGroot consensus model

The consensus model proposed by DeGroot [1974] specifies how agents in a group influence each other's opinion on the basis of mutual trust. Formally, we denote the trust of agent $i$ towards the belief of agent $j$ as $\tau_{ij} \in (0, 1]$, with $\sum_{j \in [K]} \tau_{ij} = 1$ for every $i$. Then, DeGroot defines an iterative procedure whereby each agent $i$ repeatedly updates her belief $p_i$ as

$$p_i^{(t+1)} = \sum_{j=1}^{K} \tau_{ij}\, p_j^{(t)}. \tag{1}$$

The procedure is initialized with initial beliefs $p_i^{(0)}$ and run until $p_i^{(t)} = p_j^{(t)}$ for every pair $i, j \in [K]$, at which point a consensus is deemed to have been reached. We denote the consensus belief by $p^*$.

The DeGroot model has a family resemblance to the influential PageRank algorithm [Page et al., 1999] that is designed to aggregate information across webpages in a decentralized manner. Similar to the consensus in DeGroot, the page rank corresponds to a steady state of equation (2), where the weights $\tau_{ij}$ are computed based on the fraction of outgoing links pointing from page $i$ to page $j$.

### 2.2 Computing a consensus

The DeGroot model can equivalently be thought of as specifying a weight for each agent's prediction in a linear *opinion pool* [Stone, 1961]. This suggests another way to compute the consensus prediction. Let us represent the mutual trust scores in the form of a stochastic matrix, $T = \{\tau_{ij}\}_{i,j=1}^{K}$, which defines the state transition probability matrix of a Markov chain with $K$ states. The stationary state $w$ of this Markov chain, satisfying $wT = w$, defines the relative weight of each agent's initial belief in the final consensus

$$p^* = \sum_{j=1}^{K} w_j\, p_j^{(0)}. \tag{2}$$

In our method the Markov chain is guaranteed to be irreducible and aperiodic since all entries of $T$ are positive. Therefore, a unique stationary distribution $w$ exists, and consensus will be reached from any initial state [Chatterjee and Seneta, 1977]. We refer to the weights $w$ as *consensus weights*.

To compute the consensus prediction $p^*$ one can solve the eigenvalue problem numerically for $w$ via power iteration. Letting $T^* = \lim_{m \to \infty} T^m$, the rows of $T^*$ identify, and correspond to the weights $w$ that DeGroot assigns to the individual agents [Durrett, 2019, Theorem 5.6.6]. The consensus (2) is computed as a weighted average over the initial beliefs.

## 3 DeGroot aggregation for collective test-time prediction

In this section we discuss the details of how we implement the DeGroot model outlined in Section 2.1 for defining a mechanism $\mathcal{M}$ to aggregate predictions across machine learning agents. We then analyze the resulting algorithm theoretically, proving its optimality in the large-sample limit, and we outline how the sensitivity of the procedure with respect to individual agents can be evaluated in a decentralized fashion using a Jackknife method.

---

**Algorithm 1** DeGroot Aggregation

---

1: **Input:** $K$ agents with pre-trained models $f_1, \cdots, f_K$ and local data $\mathcal{D}_k, k \in [K]$; neighborhood size $N$; test point $x'$.

2: construct trust scores (based on local accuracy):

3: **for** $i = 1, 2, \ldots, K$ **do**
4:     Construct local validation dataset $\mathcal{D}_i(x')$ using $N$-nearest neighbors of $x'$ in $\mathcal{D}_i$.
5:     Compute accuracy $\text{MSE}_{ij}(x')$ of other agents $j = 1 \ldots, K$ on $\mathcal{D}_i(x')$ according to (3).
6:     Evaluate local trust scores $\{\tau_{ij}\}_{j \in [K]}$ by normalizing $\text{MSE}_{ij}$ as in (4).
7: **end for**

8: find consensus:
9: Run pooling iterations (2), initialized at $p_j^{(0)} = f_j(x') \; \forall j$, until a consensus $p^*$ is reached.
10: **Return:** Collective prediction $p^*(x') = p^*$

---

## 3.1 Local cross-validation and mutual trust score

A key component of DeGroot's model for consensus finding is the notion of mutual trust between agents, built by information exchange in form of a discourse. Given our machine learning setup with $K$ agents each of which holds a local dataset and a model $f_k$, we simulate this discourse by allowing query access to predictions from other models. This enables each agent to validate another agent's predictive performance on its own local data. This operationalizes a proxy for trustworthiness.

An important requirement for our approach is that it should account for model heterogeneity, and define trust in light of the test sample $x'$ we want to predict. Therefore, we design an *adaptive* method, where the mutual trust $\tau_{ij}$ is reevaluated for every query via a local cross-validation procedure. Namely, agent $i$ evaluates all other agents' predictive accuracy using a subset $\mathcal{D}_i(x') \subseteq \mathcal{D}_i$ of the local data points that are most similar to the test point $x'$ [cf. Woods et al., 1997]. More precisely, agent $i$ evaluates

$$\text{MSE}_{ij}(x') = \frac{1}{|\mathcal{D}_i(x')|} \sum_{(x,y) \in \mathcal{D}_i(x')} (f_j(x) - y)^2, \tag{3}$$

locally for every agent $j \in [K]$, and then performs normalization to obtain the trust scores:

$$\tau_{ij} = \frac{1/\text{MSE}_{ij}}{\sum_{j \in [K]} 1/\text{MSE}_{ij}}. \tag{4}$$

There are various ways one can define the subset $\mathcal{D}_i(x')$; see, e.g., Cruz et al. [2018b] for a discussion of relevant distance metrics in the context of dynamic classifier selection. Since we focus on tabular data in our experiments we use Euclidean distance and assume $\mathcal{D}_i(x')$ has fixed size. Other distance metrics, or alternatively a kernel-based approach, could readily be accommodated within the DeGroot aggregation procedure.

## 3.2 Algorithm procedure and basic properties

We now describe our overall procedure. We take the trust scores from the previous section, and use them to aggregate the agents' predictions via the DeGroot consensus mechanism. See Algorithm 1 for a full description of the DeGroot aggregation procedure. In words, after each agent evaluates her trust in other agents, she repeatedly pools their updated predictions using the trust scores as relative weights. The consensus to which this procedure converges is returned as the final collective prediction, denoted $p^*(x')$. In the following we discuss some basic properties of the consensus prediction found by DeGroot aggregation.

In the context of combining expert judgement, the seminal work by Lehrer and Wagner [1981] on social choice theory has defined *unanimity* as a general property any such aggregation function should satisfy. Unanimity requires that when all agents have the same subjective opinion, the combined prediction should be no different. Algorithms 1 naturally satisfies this condition.

**Proposition 3.1** (Unanimity)**.** *If all agents agree on the prediction, then the consensus prediction from Algorithm 1 agrees with the prediction of each agent: $p^*(x') = f_i(x')$ for every $i \in [K]$.*

In addition, our algorithm naturally preserves a global ordering of the models, and the consensus weight of each agent is bounded from below and above by the minimal and maximal trust she receives from any of the other agents.

**Proposition 3.2** (Ranking-preserving). *Suppose all agent rankings have the same order: for all $j_1, j_2 \in [K]$, if $\tau_{ij_1} \geqslant \tau_{ij_2}$ for some $i \in [K]$, then $\tau_{i'j_1} \geqslant \tau_{i'j_2}$ for all $i' \in [K]$. Then, Algorithm 1 finds a consensus $p^* = \sum_i w_i f_i(x')$ where for pairs $j_1, j_2$ such that $\tau_{ij_1} \geqslant \tau_{ij_2}$, we have*

$$w_{j_1} \geqslant w_{j_2}.$$

**Proposition 3.3** (Bounded by min and max trust). *The final weight assigned to agent $i$ is between the minimal and maximal trust assigned to it by the set of agents:*

$$\max_j \tau_{ji} \geqslant w_i \geqslant \min_j \tau_{ji}.$$

Finally, Algorithm 1 recovers equally-weighted averaging whenever each agent receives a constant amount of combined total trust from the other agents. This is satisfied if all agents perform equally well on the different validation sets. However, the trust not necessarily needs to be allocated uniformly, there may be multiple $T$ that result in a uniform stationary distribution.

**Proposition 3.4** (Averaging as a special case). *If the columns of $T$ sum to 1, then Algorithm 1 returns an equal weighting of the agents: $w_i = 1/K$ for $i = 1, \ldots, K$.*

Together these properties serve as a basic sanity check that Algorithm 1 implements a reasonable consensus-finding procedure, from a decision-theoretic as well as algorithmic perspective.

### 3.3 Optimality in a large-sample limit

In this section we analyze the large-sample behavior of the DeGroot consensus mechanism, showing that it recovers the optimal weighting scheme in the scenario where agents are independent.

For our large-sample analysis, we suppose that the agents' local datasets $\mathcal{D}_k$ are drawn independently from (different) distributions $\mathcal{P}_k$ over $\mathcal{X} \times \mathcal{Y}$. In order for our prediction task to be well-defined, we assume that the conditional distribution of $Y$ given $X$ is the same for all $\mathcal{P}_k$. In other words, the distributions $\mathcal{P}_k$ are all covariate shifts of each other. For simplicity, we assume the distributions $\mathcal{P}_k$ are all continuous and have the same support. We will consider the large-sample regime in which each agent has a growing amount of data, growing at the same rate. That is, if $n = |\mathcal{D}_1| + \cdots + |\mathcal{D}_k|$ is the number of total training points, then $|\mathcal{D}_k|/n \to c_k > 0$, as $n \to \infty$ for each agent $k = 1, \ldots, K$. Lastly, we require a basic consistency condition: for any compact set $\mathcal{A} \subset \mathcal{X}$, $f_k \to f_k^*$ uniformly over $x \in \mathcal{A}$ as $n \to \infty$, for some continuous function $f_k^*$.

**Theorem 3.5** (DeGroot converges to inverse-MSE weighting). *Let $x' \in \mathcal{X}$ be some test point. Assume that $\mathcal{P}_k$ is supported in some ball of radius $\delta_0$ centered at $x'$, and that the first four conditional moments of $Y$ given $X = x$ are continuous functions of $x$ in this neighborhood. Next, suppose we run Algorithm 1 choosing $N = \lceil n^c \rceil$ nearest neighbors for some $c \in (0, 1)$. Let*

$$\text{MSE}_k^* = \mathbb{E}\left[(Y - f_k^*(x'))^2\right]$$

*denote the asymptotic MSE of model $k$ at the test point $x'$, where the expectation is over a draw of $Y$ from the distribution of $Y \mid X = x'$. Then, the DeGroot consensus mechanism yields weights*

$$w_k \to w_k^* = \frac{1/\text{MSE}_k^*}{1/\text{MSE}_1^* + \cdots + 1/\text{MSE}_K^*}.$$

In other words, in the large-sample limit the DeGroot aggregation yields inverse-MSE weighting and thus recovers inverse-variance weighting for unbiased estimators—a canonical weighting scheme from classical statistics [Hartung et al., 2008] that is known to be the optimal way to linearly combine independent, unbiased estimates. This can be extended to our setting to show that DeGroot averaging leads to the optimal aggregation when we have independent[1] agents, as stated next.

---

[1]While independence is a condition that we can't readily check in practice, it may hold approximately if the agents use different models constructed from independent data. In any case, the primary takeaway is that DeGroot is acting reasonably in this tractable case.

**Theorem 3.6** (DeGroot is optimal for independent, unbiased agents). *Assume that the conditional mean and variance of $Y$ given $X = x$ are continuous functions of $x$. For some $\delta > 0$, consider drawing $\tilde{X}$ uniformly from a ball of radius $\delta$ centered at $x'$ and $\tilde{Y}$ from the distribution of $Y \mid X = \tilde{X}$. Under this sampling distribution, suppose the residuals $\tilde{Y} - f_k^*(\tilde{X})$ from the agents' predictions have mean zero and each pair has correlation zero. Then the optimal weights,*

$$\tilde{w} := \underset{w \in \mathbb{R}^k : \|w\|_1 = 1}{\arg\min} \mathbb{E}_{(\tilde{X}, \tilde{Y})} \left[ \left( \tilde{Y} - \sum_{k \in [K]} w_k f_k^*(\tilde{X}) \right)^2 \right],$$

*approach the DeGroot weights: $\tilde{w} \to w^*$ as $\delta \to 0$.*

In summary, Theorem 3.5 shows that the DeGroot weights are asymptotically converging to a meaningful (sometimes provably optimal) aggregation of the models – *locally for each test point $x'$*. This is a stronger notion of adaptivity than that usually considered in the model-averaging literature.

### 3.4   Error bars for predictions

Finally, we develop a decentralized Jackknife algorithm [Quenouille, 1949, Efron and Tibshirani, 1993] to estimate the standard error of the consensus prediction $p^*(x')$. Our proposed procedure measures the impact of excluding a random agent from the ensemble and returns error bars that measure how stable the consensus prediction is to the observed collection of agents. Formally, let $p_{-i}^*(x')$ denote the consensus reached by the ensemble after removing agent $i$. Then, the Jackknife estimate of standard error at $x'$ corresponds to

$$\widehat{\text{SE}}(x') = \sqrt{\frac{K-1}{K} \sum_{i \in [K]} \left( p_{-i}^*(x') - \bar{p}^*(x') \right)^2}, \tag{5}$$

where $\bar{p}^*(x') = \frac{1}{K} \sum_{i=1}^{K} p_{-i}^*(x')$ is the average delete-one prediction.

In the collective prediction setting, this is an attractive target of inference because it can be computed in a decentralized manner, is entirely agnostic to the data collection or training mechanism employed by the agents, and requires no additional model evaluations above those already carried out in Algorithm 1. Furthermore, it allows one to diagnose the impact of a single agent on the collective prediction and thus assess the vulnerability of the algorithm to malicious agents. A detailed description of the DeGroot Jackknife procedure can be found in Algorithm 2 in Appendix B. Finally, we want to note that this measure is not specific to how the consensus is reached via the DeGroot method and can readily be applied as a diagnosis tool to other collective prediction algorithms.

### 3.5   Privacy-preserving properties

It is an appealing property of the DeGroot algorithm that it does not require agents to share their raw training data or model parameters upfront. Information is only exchanged as needed through requesting and answering prediction queries of other agents. This implies that agents can do privacy accounting and remain in control of their data at any time. Sensitive information about the training data can only by leaked in Step 5 of Algorithm 1 by revealing unlabeled data points and prediction queries. For privacy-critical applications this information could further be protected by augmenting DeGroot aggregation with techniques from differential privacy [Dwork and Roth, 2014], or tools from cryptography [Bonawitz et al., 2017]. While we are not providing a rigorous privacy analysis of our method in this work, it could be an interesting extension for future work.

## 4   Experiments

We investigate the efficacy of the DeGroot aggregation mechanism empirically for various datasets, partitioning schemes and model configurations. We start with a synthetic setup to illustrate the strengths and limitations of our method. Then, we focus on real datasets to see how these gains surface in more practical applications with natural challenges, including data scarcity, heterogeneity of different kinds, and scalability. We compare our Algorithm 1 (DeGroot) to the natural baseline of equally-weighted model averaging (M-avg) and also compare to the performance of individual

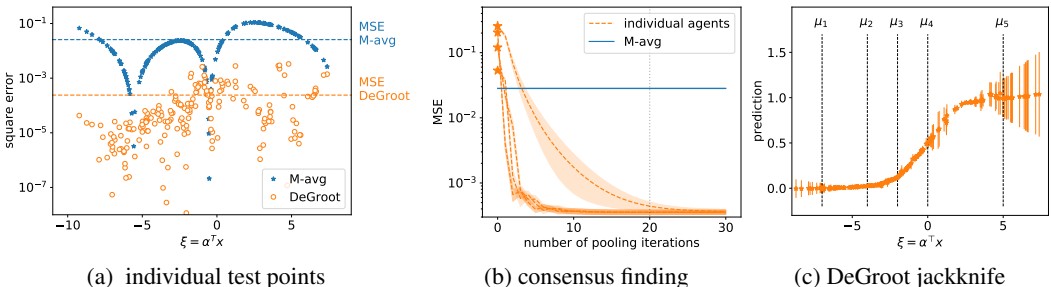

|     |     |     |
|:---:|:---:|:---:|
| (a) individual test points | (b) consensus finding | (c) DeGroot jackknife |

Figure 2: *(Synthetic data experiment)*. Comparison of DeGroot to M-avg. (a) Predictive accuracy on individual test points. (b) Performance for varying number of pooling iterations during DeGroot consensus finding. (c) Confidence intervals for individual test points returned by DeGroot Jackknife.

models composing the ensemble. In Section 4.2 we include an additional comparison with two reference schemes that have access to shared validation data and a central agent that determines an (adaptive) weighting scheme—information DeGroot does not have.

## 4.1 Synthetic data

First, we investigate a synthetic two-dimensional setting where each agent's features are drawn from a multivariate Gaussian distribution, $x \sim \mathcal{N}(\mu_k, \Sigma_k)$. The true labeling function is given as $y = [1 + \exp(\alpha^\top x)]^{-1}$ and we add white noise of variance $\sigma_Y^2$ to the labels in the training data. Unless stated otherwise, we use $K = 5$ agents and let each agent fit a linear model to her local data. See Figure 4 in the Appendix for a visualization of the models learned by the individual agents, and how they accurately approximate the true labeling function in different regions of the input space.[2]

To evaluate the DeGroot aggregation procedure we randomly sample 200 test points from the uniform mixture distribution spanned by the $K$ training data distributions, and compare the prediction of DeGroot to M-avg. We visualize the squared error for the collective prediction on the individual test points in Figure 2a where we use $\xi := \alpha^\top x$ as a summary for the location of the test point on the logistic curve. We observe that DeGroot consistently outperforms M-avg across the entire range of the feature space. Exceptions are three singularities, where the errors of the individual agents happen to cancel out by averaging. Overall DeGroot achieves an MSE of $4.6e^{-4}$ which is an impressive reduction of over $50\times$ compared to M-avg. Not shown in the figure is the performance of the best individual model which achieves an MSE of $5e^{-2}$ and performs worse than the M-avg baseline. Thus, DeGroot outperforms an oracle model selection algorithm and every agent strictly benefits from participating in the ensemble. The power of adaptive weighting such as used in DeGroot is that it can trace out a nonlinear function, given only linear models, whereas any static weighting (or model selection) scheme will always be bound to the best linear fit.

In Figure 2b we show how individual agent's predictions improve with the number of pooling iterations performed in Step 9 of Algorithm 1. The influence of each local model on the consensus prediction of $x'$ and how it changes depending on the location of the test point is illustrated in Figure 1 in the introduction.

An interesting finding that we expand on in Appendix D.1 is that for the given setup the iterative approach of DeGroot is superior to a more naive approach of combining the individual trust scores or local MSE values into a single weight vector. In particular, DeGroot reduces the MSE by $20\times$ compared to using the average trust scores as weights. A theoretical understanding of this remarkable phenomenon remains to be developed.

In Figure 2c we visualize the error bars returned by our DeGroot Jackknife procedure. As expected, the intervals are large in regions where only one a small number of agents possess representative training data, and small in regions where there is more redundancy across agents.

---

[2]If not stated otherwise we spread the means as $\mu = \{[-3,-4], [-2,-2], [-1,-1], [0,0], [3,2]\}$ and let $\Sigma_k = I$. We use 200 training samples on each agent, for the labeling function we use $\alpha = [1, 1]$, and for the label noise in the training data we set $\sigma_Y = 0.1$. We use $N = 5$ for local cross-validation in DeGroot.

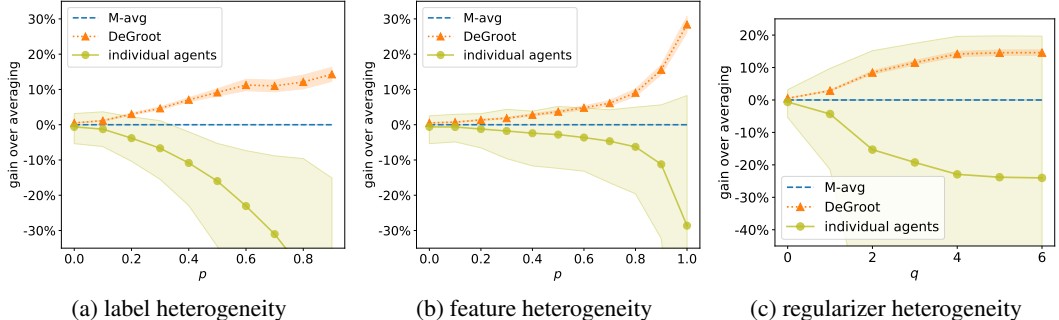

Figure 3: *(Data and model heterogeneity for the abalone data).* Relative gain over M-avg. Confidence intervals of DeGroot are over randomness in data partitioning and for the individual agents the shaded area spans the performance from the worst to the best model, the line indicating average performance.

Finally, in Appendix D.1 we conduct further qualitative investigations of the DeGroot algorithm. First, we vary the covariance in the local data feature distribution. We find that the gain of DeGroot over M-avg becomes smaller as the covariance is increased; models become less specialized and there is little for DeGroot to exploit. On the other extreme, if the variance in the data is extremely small, the adaptivity mechanism of DeGroot becomes less effective, as the quality of individual models can no longer be reliably assessed. However, we need to go to extreme values in order to observe these phenomena. In a second experiment we explore DeGroot's sensitivity to choices of $N$. We find that when $N$ is chosen too large adaptivity can suffer, and for very small values of $N$ performance can degrade if the labels are highly noisy. However, there is a broad spectrum of values for which the algorithm performs well, between $1\%$ and $10\%$ of the local data usually a good choice.

## 4.2 Real datasets

Turning to real data sets, we first investigate how DeGroot deals with three different sources of heterogeneity. We work with the abalone dataset [Nash et al., 1994] and train a lasso model on each agent[3] with regularization parameter $\lambda_k = \lambda'$ that achieves a sparsity of ∼0.8. For our first experiment, shown in Figure 3a, we follow a popular setup from federated learning to control heterogeneity in the data partitioning [see, e.g., Shen et al., 2021]. We partition the training data in the following manner: a fraction $1 - p$ of the training data is partitioned randomly, and a fraction $p$ is first sorted by the outcome variable $y$ and then partitioned sequentially. For the second experiment, shown in Figure 3b, we use a similar partitioning scheme, but sort the data along an important feature dimension, instead of $y$. This directly leads to heterogeneity in the input domain. Finally, we investigate model heterogeneity arising from different hyper-parameter choices of the regularization parameter $\lambda$, given a random partitioning. To control the level of heterogeneity, we start with $\lambda = \lambda'$ on each agent, and let the parameters diverge increasingly, such that $\lambda_k = \lambda' \left[1 + \frac{k-k_0}{K}\right]^q$ with $k_0 = 3$. The results are depicted in Figure 3c. In all three experiments the gain of DeGroot over M-avg is more pronounced with increasing degree of heterogeneity. When there is heterogeneity in the partitioning, DeGroot clearly outperforms the best individual model, by taking advantage of models' local areas of expertise. For the third experiment where the difference is in model quality, DeGroot manages to almost recover the performance of the best single model.

Next, we conduct a broad comparison of DeGroot to additional reference schemes across various datasets and models, reporting the results in Table 1. To the best of our knowledge there is no existing generic adaptive aggregation scheme that does not rely on external validation data to either build the aggregation function or determining the weights $w$. Thus, we decided to include a comparison with an approach that uses a validation dataset to compute the weighting, where the external validation data has the same size as the local partitions. Our two references are static inverse-MSE weighting (CV-static) where the weights are determined based on the model's average performance on the validation data, and adaptive inverse-MSE weighting (CV-adaptive), where the performance is evaluated only in the region around the current test point $x'$. We achieve heterogeneity across local datasets by

---

[3]Similar phenomena can be observed for other datasets and models (see Appendix D.2)

Table 1: *(Benchmark comparison).* Performance for different dataset/model combinations. Datasets have been downloaded from [Fan, 2011]. We use ridge and lasso as a baseline and train a decision tree regressor (DTR) for discrete outcome variables, and a neural net (NN) with two hidden layers for continuous outcomes. A positive relative gain means a reduction in MSE compared to DeGroot.

| | | MSE DeGroot | gain relative to DeGroot [%] | | | hyper-parameters |
|---|---|---|---|---|---|---|
| | | | M-avg | CV-static | CV-adaptive | |
| Boston | Ridge | 25.23 | -12.45±2.29 | -10.24±1.90 | -2.80±1.72 | $\lambda = 1e^{-5}$ |
| | Lasso | 24.17 | -13.70±1.72 | -10.35±1.16 | -3.46±1.44 | $\lambda = 5e^{-3}$ |
| | NN | 14.18 | -15.18±3.43 | -11.01±3.08 | -5.81±3.20 | layer= $(7, 7)$ |
| E2006 | Ridge | 0.15 | -0.12±0.06 | -0.21±0.06 | 0.13±0.11 | $\lambda = 5e^{-2}$ |
| | Lasso | 0.097 | -0.92±0.15 | -0.90±0.13 | -0.08± 0.07 | $\lambda = 5e^{-5}$ |
| | NN | 0.11 | -14.58±3.48 | -5.69±0.97 | -0.16±0.53 | layer= $(9, 9)$ |
| Abalone | Ridge | 3.18 | -3.67 ± 0.55 | -3.52±0.47 | -0.49±0.46 | $\lambda = 5e^{-2}$ |
| | Lasso | 4.93 | -10.09± 0.83 | -10.05±0.80 | -0.39±0.75 | $\lambda = 5e^{-2}$ |
| | DTR | 4.85 | -2.53± 0.69 | -2.55±0.70 | -0.79±0.88 | max_depth = 4 |
| cpusmall | Ridge | 59.00 | -96.23±23.46 | -89.81±15.58 | -0.78±0.63 | $\lambda = 1e^{-5}$ |
| | Lasso | 53.82 | -76.08±14.45 | -73.81±10.51 | -2.04±1.82 | $\lambda = 1e^{-3}$ |
| | DTR | 11.16 | -4.05±1.88 | -3.90±1.81 | 1.65±2.20 | max_depth=7 |
| YearPrediction | Ridge | 95.02 | -0.63±0.24 | -0.94±0.11 | 0.21±0.12 | $\lambda = 1$ |
| | Lasso | 91.21 | -1.73±0.47 | -1.98±0.35 | -0.22±0.29 | $\lambda = 1$ |
| | DTR | 62.11 | -3.13±1.83 | -2.26±1.02 | -0.54±0.53 | max_depth = 4 |

partially sorting a fraction $p = 0.5$ of the labels, and we choose $N$ to be $1\%$ of the data partition for all schemes (with a had lower bound at 2). Each scheme is evaluated for the same set of local models and confidence intervals are over the randomness of the data splitting into partitions, test and validation data across evaluation runs. The results demonstrate that DeGroot can effectively take advantage of the proprietary data to find a good aggregate prediction, significantly outperforming M-avg and CV-static, and achieving better or comparable performance to CV-adaptive that works with additional data.

To complement our study, we verify in Appendix D.2 that our aggregation mechanism scales robustly with the number of agents in the ensemble and the simultaneously decreasing partitioning size: DeGroot consistently outperforms model averaging on two different datasets on the full range from 2 up to 128 agents. Finally, for the reader who is interested in applying the DeGroot procedure to classification tasks we have outlined a classification example in Appendix D.3.

## 5    Discussion

We have shown that insights from the literature on humans consensus via discourse suggest an effective aggregation scheme for machine learning. In settings where there is no additional validation data available for assessing individual models' quality, our approach offers an appealing mechanism to take advantage of individual agents' proprietary data to arrive at an accurate collective prediction. Our mechanism relies on information exchange between agents formalized through a query model. This emulation of a decentralized discussion phase that allows agents to influence each other's opinion offers an appealing new perspective on decentralized learning. We are convinced that building on this idea, developing richer models of discourse, and refining mutual trust through experience could offer a promising avenue for future research in federated learning.

## Acknowledgments and funding statement

The authors would like to thank Jacob Steinhardt and Alex Wei for feedback on this manuscript. In terms of funding; CM was supported by the Swiss National Science Foundation Early Postdoc Mobility fellowship program, WG was supported by a Google PhD fellowship, and MJ acknowledges support from the Vannevar Bush Faculty fellowship program under grant number N00014-21-1-2941.

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
