# A  Proofs

For some of our proofs we will use a Markov chain interpretation of the DeGroot mechanism, where the matrix $T$ defines the transition probabilities of the Markov chain. By definition of the trust scores, the rows of $T$ sum to one. Moreover, the entries are all positive, so this is an aperiodic, irreducible Markov chain and hence has a unique stationary distribution, given by the solution $w$ to the linear system $w = Tw$ [e.g., Häggström, 2002]. Furthermore, $T$ has a unique largest left-eigenvector with eigenvalue 1. It will be convenient for us to consider a Markov chain $Z_0, Z_1, \ldots$ with transition matrix $T$, and $Z_0$ drawn from the stationary distribution $w$.

*Proof of Proposition 3.1.* We prove the claim by induction on the number of consensus steps $t$. First, we have that $p_i^{(0)} = f_1(x')$ for all $i \in [K]$ by assumption. Suppose $p_i^{(t)} = f_1(x')$ for all $i \in [K]$. Then,

$$p_i^{(t+1)} = \sum_{j=1}^{K} \tau_{i,j} p_j^{(t)} = f_1(x')$$

since the weights sum to 1. ∎

*Proof of Proposition 3.2.* Using our Markov chain notation above, it suffices to show that

$$P(Z_1 = j_1) \geqslant P(Z_1 = j_2).$$

It is easy to see that this is the case since

$$P(Z_1 = j_1 \mid Z_0 = i) \geqslant P(Z_1 = j_2 \mid Z_0 = i)$$

for all $i \in [K]$. ∎

*Proof of Proposition 3.3.* Using our Markov chain notation above, note that

$$\max_{j}\{P(Z_1 = i \mid Z_0 = j)\} \geqslant P(Z_1 = i) \geqslant \max_{j}\{P(Z_1 = i \mid Z_0 = j)\}.$$

The result follows. ∎

*Proof of Proposition 3.4.* Note that in this case, $1^\top T = 1$ (where 1 is the $K$-vector of all ones), so $w = 1/n$ is the stationary state. ∎

*Proof of Theorem 3.5.* Choose any $i \in [K]$. We will show that $\tau_{ij} \to 1/\mathrm{MSE}_j^*$ in probability as $n \to \infty$.

First, let $x_1, \ldots, x_N \in \mathcal{D}_i$ be the $N$ nearest neighbors to the test point $x'$ in agent $i$'s data. We claim that

$$\max_{m \in \{1, \ldots, N\}} \|x_m - x'\| \to 0 \tag{6}$$

in probability as $n \to \infty$. To see this, for $\delta_0 > \delta > 0$, note that $P(\|x_m - x'\| < \delta) = \varepsilon$ for some $\varepsilon > 0$, since the points in $\mathcal{D}_i$ are an i.i.d. sample from a distribution supported in this ball. But then,

$$P\left(\max_{m \in \{1, \ldots, N\}} \|x_m - x'\| \geqslant \delta\right) = P(B < N)$$

where $B$ is distributed as a binomial with $|\mathcal{D}_i|$ draws and success probability $\varepsilon$. But

$$P\left(B < N\right) = P\left(\frac{B - \varepsilon|\mathcal{D}_i|}{\sqrt{|\mathcal{D}_i|\varepsilon(1 - \varepsilon)}} < \frac{N - \varepsilon|\mathcal{D}_i|}{\sqrt{|\mathcal{D}_i|\varepsilon(1 - \varepsilon)}}\right)$$

By assumption the term $N - \varepsilon|D_i|/\sqrt{|\mathcal{D}_i|\varepsilon(1 - \varepsilon)} \to -\infty$ as $n \to \infty$ since $N/|\mathcal{D}_i| \to 0$. Then, by the central limit theorem we conclude that

$$P\left(B < N\right) \to 0$$

as $n \to \infty$, and hence (6) holds.

Next, let $\mu(x) = \mathbb{E}[Y \mid X = x]$ and $\sigma(x) = \mathrm{Var}(Y \mid X = x)$ be the true mean and variance functions, respectively. For convenience, let $B_\delta(x')$ be the ball of radius $\delta$ centered at $x'$. We have

$$\inf_{x \in B_\delta(x')} \{\mu(x) - f_j(x))^2 + \sigma(x)^2\} \leqslant \mathbb{E}\left[1/\tau_{i,j}\big|\max_{m \in \{1,\ldots,N\}} \|x_m - x'\| < \delta\right] \leqslant \sup_{x \in B_\delta(x')} \{\mu(x) - f_j(x))^2 + \sigma(x)^2\}.$$

Note that the lower and upper bound both converge to $\mathrm{MSE}_j^*$ as $\delta \to 0$, by the continuity of $\mu(x), f_j(x)$ and $\sigma(x)$. Similarly, the conditional variance of $1/\tau_{i,j}$ is bounded by $C/N$ for some constant $C$. Thus, by Chebyshev's inequality, there exist sequences $c_1(n) \to 0$ and $c_2(n) \to 0$ such that

$$P\left(|\tau_{ij} - \mathrm{MSE}_j^*| \geqslant c_1(n) \mid \max_{m \in \{1,\ldots,N\}} \|x_m - x'\| < \delta\right) \leqslant c_2(n).$$

Since the event $\{\max_{m \in \{1,\ldots,N\}} \|x_m - x'\| < \delta\}$ has probability tending to 1 as $n$ grows, this implies that $1/\tau_{i,j} \to \mathrm{MSE}_j^*$ in probability, as desired. ∎

*Proof of Theorem 3.6.* For this distribution, we wish to find weights $\tilde{w}$ solving the following:

$$\tilde{w} := \operatorname*{arg\,min}_{w \in \mathbb{R}^k, \mathbf{1}^\top w = 1} \mathbb{E}_{(\tilde{X},\tilde{Y})}\left[\left(\tilde{Y} - \sum_{k=1}^{K} w_k f_k^*(\tilde{X})\right)^2\right].$$

Expanding the right side, we have

$$\mathbb{E}_{(\tilde{X},\tilde{Y})}\left[\left(\tilde{Y} - \sum_{k=1}^{K} w_k f_k^*(\tilde{X})\right)^2\right] = \mathbb{E}_{(\tilde{X},\tilde{Y})}\left[\left(\sum_{k=1}^{K} w_k (\tilde{Y} - f_k^*(\tilde{X}))\right)^2\right]$$

$$= \mathrm{Var}\left(\sum_{k=1}^{K} w_k (\tilde{Y} - f_k^*(\tilde{X}))\right)$$

$$= \sum_{k=1}^{K} \mathrm{Var}\left(w_k (\tilde{Y} - f_k^*(\tilde{X}))\right)$$

$$+ 2 \sum_{1 \leqslant k_1 < k_2 \leqslant K} \mathrm{Cov}\left(w_{k_1}(\tilde{Y} - f_{k_1}^*(\tilde{X})), w_{k_2}(\tilde{Y} - f_{k_2}^*(\tilde{X}))\right)$$

$$= \sum_{k=1}^{K} w_k^2 \mathrm{Var}\left(\tilde{Y} - f_k^*(\tilde{X})\right)$$

With this expression, one can easily verify that the optimal $w$ is then

$$\tilde{w}_k = 1/\mathrm{Var}\left(\tilde{Y} - f_k^*(\tilde{X})\right).$$

Lastly, we claim that

$$\mathrm{Var}\left(\tilde{Y} - f_k^*(\tilde{X})\right) \to \mathrm{MSE}_k^*$$

as $\delta \to 0$. This follows from the decomposition

$$\mathrm{Var}\left(\tilde{Y} - f_k^*(\tilde{X})\right) = \mathbb{E}\left[\mathrm{Var}\left(\tilde{Y} - f_k^*(\tilde{X}) \mid \tilde{X}\right)\right] + \mathrm{Var}\left(\mathbb{E}[\tilde{Y} - f_k^*(\tilde{X}) \mid \tilde{X}]\right).$$

By assumption, the conditional mean and variance are continuous functions of $x$, so the first term in the sum converges to $\mathrm{MSE}_k^*$ as $\delta \to 0$. Likewise, the second term in the sum converges to 0. This completes the proof.

∎

## B   Estimating the Standard Error

In this section, we develop a standard errors estimator for the collective prediction $p^*(x')$ resulting from Algorithm 1. Our proposal is a special case of jackknife standard error estimates, where we consider the agents to be the independent samples from a super-population of agents (this probabilistic interpretation is useful to precisely discuss standard errors but is not needed elsewhere in our work). Our error bars will thus measure how stable the consensus prediction is to the observed collection of agents. In the collective prediction setting, this is an attractive target of inference because it can be estimated without requiring any assumptions about how the agents gather data, their models and training procedure. As a result, our standard error estimate is entirely agnostic to the behavior of the agents, is decentralized, and requires minimal communication—it requires no additional model queries above what was already done in Algorithm 1.

---

**Algorithm 2** DeGroot Jackknife

---

1: **Input:** Pre-trained $K$ agents $f_1, \cdots, f_K$ with local training dataset $\mathcal{D}_k, k \in [K]$; neighborhood size $m$; test point $x'$.

2: Construct trust matrix $T$ as in Algorithm 1.

3: **for** $i = 1, 2, \ldots, K$ **do**
4:     Create submatrix $T^{(i)}$
5:     Find $v$ such that $vT^{(i)} = v$ by power iteration.
6:     Form collective prediction $p^*_{-i}(x') = \sum_{j \neq i} v_j f_j(x')$
7: **end for**

8: **Return:** Jackknife estimate of standard error at $x'$:

$$\widehat{\mathrm{SE}}(x') = \sqrt{\frac{K-1}{K} \sum_{i=1}^{K} \left( p^*_{-i}(x') - \bar{p}^*(x') \right)^2},$$

    where $\bar{p}^*(x') = \frac{1}{K} \sum_{i=1}^{K} p^*_{-i}(x')$ is the average delete-one prediction at $x'$.

---

Turning to the details, recall that $T$ is the matrix of trust scores, and let $T^{(i)}$ be the principal submatrix of $T$ formed by deleting row $i$ and column $i$, renormalized so that each row sums to one. That is, $T^{(i)}$ is the trust matrix if agent $i$ is removed from our collection of agents. The idea is that we run DeGroot aggregation with agent $i$ deleted; i.e., we find the collective prediction $p^*_{-i}(x')$ of the remaining $K-1$ agents. Then, by looking at the variability of these predictions with different agents removed, we can quantify the stability of the procedure. Specifically, we take the sample standard deviation of these quantities, scaled by $\sqrt{(K-1)^2/K}$: the scaling prescribed by the theory of the jackknife estimator [Efron and Tibshirani, 1993]. We state this procedure formally in Algorithm 2.

Importantly, this calculation does not require any additional model evaluations above what is required to perform the consensus prediction; this algorithm only requires constructing the matrix $T$ and knowing the predictions $f_j(x')$, which were already needed in Algorithm 1. Thus, estimating the standard error has no additional information-sharing requirement; it requires only a modest extra amount of computation.

See Figure 8 for an empirical demonstration of these error bars on our synthetic example from Section 4.1.

## C   Details on algorithm implementation

In this section we discuss some practical considerations when implementing Algorithm 1.

*Step 4.* For our experiments we have chosen the Euclidean distance metric, and a fixed number $N$ of datapoints to determine $\mathcal{D}_i(x')$. To find the $N$ nearest neighbors we have used the `sklearn` unsupervised `NearestNeighbors` classifier from `sklearn.neighbors`. Note that this step could be adjusted to use other distance metrics specific to the modality of the data. In particular, for higher dimensional data such as images a careful choice of distance metric will become crucial.

*Step 9.* In this step the agents iteratively refine their predictions by pooling other agents predictions to reach a consensus, as described in Section 2.1. These pooling iterations can be executed in a fully decentralized manner, or the consensus weights can be evaluated via the power method at one of the nodes that gets access to all the trust scores in order to construct the matrix $T$ from $\{\tau_{i,j}\}_{i,j\in[K]}$. The best implementation will depend on the requirements of your application. For our experiments we have chosen the centralized evaluation for convenience. Around $t_p = 30$ power iterations are usually far sufficient to reach a consensus. Given that the number of agents is usually small ($< 1000$) this operation comes with little computational cost and we recommend to pick the number of iterations $t_p$ with sufficient margin to take advantage of the full potential of DeGroot aggregation. Alternatively, one might also implement a stopping criteria in the form of $|p_j^{(t)} - p_j^{(t-1)}| \leqslant$ tol that is checked independently by each agent $j \in [K]$ after every round.

*Step 10.* For every test point the algorithm returns the consensus prediction $p^*(x')$. If Step 9 has fully converged we have $p_j = p^*$ for all $j$ and we can return any of the individual agents predictions. However, to get a robust procedure, even if the individual beliefs have not fully converged, we return $p^* = \frac{1}{K} \sum_j p_j$ instead.

# D    Additional Experimental Details and Evaluations

The code is written in Python and for model training, as well as the nearest neighbor computation we use the built-in functionalities of Sklearn [Pedregosa et al., 2011].

## D.1    Evaluation on Synthetic Data

For the synthetic setup the true label is determined by a logistic function, as described in Section 4.1, where we choose $\alpha = [1, 1]$. The features of agent $k$ are distributed according to a multi-variate Gaussian distribution with mean $\mu_k$ and each model then fits a linear function to it's training data. We use the following default experimental configuration: we choose the means as $\mu_1 = [-3, -4]$, $\mu_2 = [-2, -2]$, $\mu_3 = [-1, -1]$, $\mu_4 = [0, 0]$ $\mu_5 = [3, 2]$, the covariance of the local data as $\Sigma_k = \Sigma = I$ and the variance in the label noise as $\sigma_Y = 0.1$. The resulting local linear fits are illustrated in Figure 4 where we see that individual models approximate the true labeling function in different regions of the input space. The accuracy as a function of $\xi$ is illustrated in Figure 1 in the introduction.

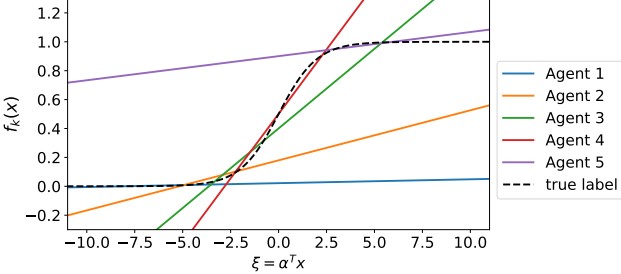

Figure 4: *(agents' local models).*  Local linear models fit by individual agents for the default experimental configuration.

**Additional Evaluations:**

*A) Sensitivity of DeGroot to hyperparameter $N$.* In Figure 5 we demonstrate how the performance of Algorithm 1 in this synthetic setting changes with the number of neighbors $N$ used for local validation for two different levels of label noise. Overall we find that the performance of DeGroot is not very sensitive to the choice of $N$, whereas the optimal regime of values for $N$ increases with the noise in the data. In general, we recommend choosing $N$ to corresponds to approximately $1 - 10\%$ of the available data, with a hard lower-bound on $N$, bounding it away from 1. For later investigations on real data we pick $N$ to be $1\%$ of the partition size.

*B) Varying overlap in local data partitions.* In Figure 6 we investigate how the performance of DeGroot changes in comparison to M-avg for varying overlap in the local data partitions. Therefore

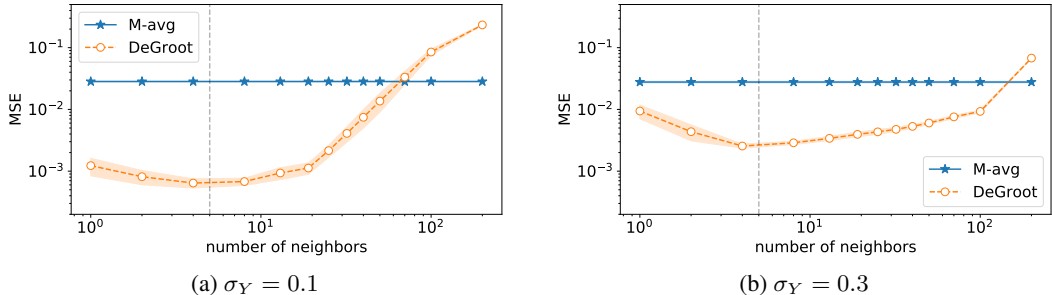

(a) $\sigma_Y = 0.1$             (b) $\sigma_Y = 0.3$

Figure 5: *(Performance vs. number of neighbors).* Synthetic data experiment comparing DeGroot to model averaging (M-avg) for two different noise levels ($\sigma_Y$) in the training data labels. Gray dashed line indicate the default value in our experiments.

we vary the covariance $\Sigma_k = I\sigma^2$ in the feature distribution of the individual agents, while keeping the means $\mu_k$ fixed. We find that for larger values of $\sigma^2$ there is less to gain for DeGroot since models are less specialized, although the gains remains significant up to $\sigma^2 = 10$. We further see that if the variance in the data is very small and the local datasets are not sufficiently covering the feature space, the local cross-validation procedure and the adaptivity of the DeGroot procedure start to suffer.

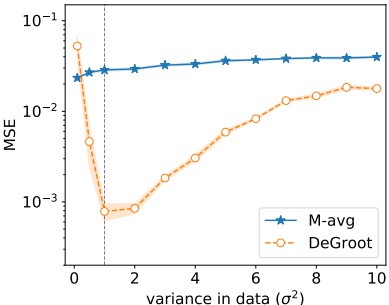

Figure 6: *(Varying overlap in data partitions).* Comparing DeGroot to classical model averaging (M-avg) for varying covariance $\Sigma_k = I\sigma^2$ in the feature distribution, keeping the means $\mu_k$ fixed. Gray dashed line indicate the default value in our experiments.

*C) Alternative pooling operations.* In Figure 7 we aim to provide additional insights into the effect of DeGroot's iterative procedure to find consensus and aggregate the individual predictions. In Figure 2b we have shown how the accuracy of individual agents improves with the number of pooling iterations in the DeGroot procedure. After around 20 iterations the agents have reached consensus. In Figure 7 we compare the accuracy of the predictions obtained through DeGroot to alternative ways of aggregating the local MSE values and trust scores into a single weight vector. These baselines are constructed for diagnostic purpose to demonstrate the value of using power iterations instead of an alternative procedure.

The first method, denoted $\tau$-avg takes the mean of the trust scores $\tau_{ij}$ across all agents $i$ to obtain the weights $w_j$ of agent $j$ on the final prediction. As shown in Figure 7a $\tau$-avg performs significantly worse than DeGroot on our synthetic example, and achieves an overall MSE of $9e^{-3}$ which is $20\times$ larger than DeGroot. This shows that the DeGroot agents find a better collective prediction through iterative pooling than they would by averaging their beliefs after only a single round of pooling.

The second method first averages the local MSE values of each agent across the different datasets as $\overline{\text{MSE}}_j = \sum_i \text{MSE}_{i,j}$ and then obtains the weights through inverse-MSE weighting as $w_j = 1/\overline{\text{MSE}}_j$. Similar to $\tau$-avg, this alternative procedure performs significantly worse than DeGroot, and acheives an MSE of $1e^{-2}$.

*D) DeGroot Jackknife.* Finally, in Figure 8, we evaluate the error bars for our predictions using the Jackknife procedure proposed in Algorithm 2 for the test points evaluated in Figure 2a. Here, we

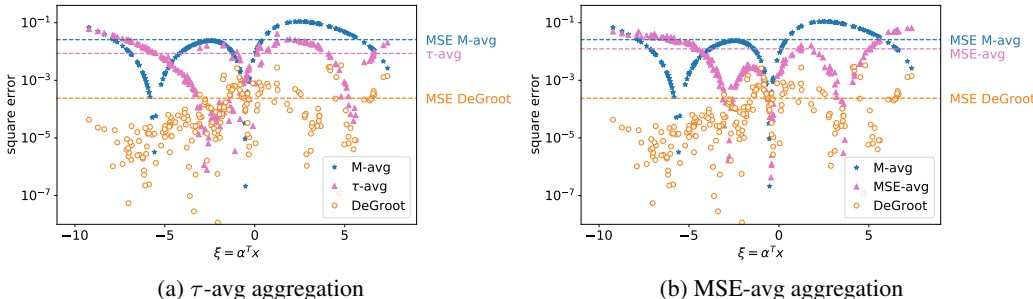

(a) $\tau$-avg aggregation          (b) MSE-avg aggregation

Figure 7: *(Alternative aggregation schemes).* (a) Comparison of DeGroot to $\tau$-avg that average the trust scores across all agents to obtain the weights of the individual agents. (b) Comparison of DeGroot to MSE-avg that performs inverse-MSE weighting based on the aggregated MSE values cross the local datasets. Experiments serve for diagnosis purpose and are conducted on the synthetic setup outlined in Section 4.1.

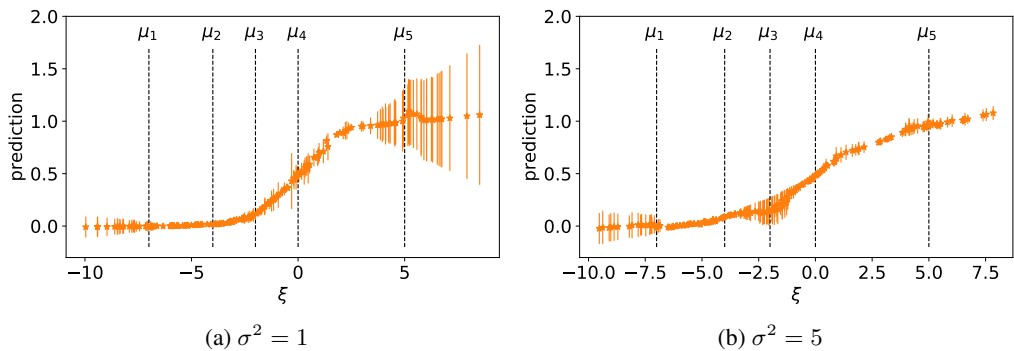

(a) $\sigma^2 = 1$          (b) $\sigma^2 = 5$

Figure 8: *(DeGroot Jackknife).* Evaluations of error bars using the decentralized Jackknife procedure proposed in Algorithm 2. The test points correspond to the same 200 test points visualized in Figure 2a. For (a) we use the standard configuration, and for (b) we increase the covariance of the local training datasets by a factor of 5, i.e. $\Sigma_k = 5\,\mathrm{I}$.

report the predictions as a function of $\xi$, as well as the standard error (5). We find that the standard errors are relatively small in the center, but increase at the edges of the space. This makes sense: on the edges of the space there is only one agent with good prediction accuracy, so the final consensus prediction is not stable to the deletion of any one agent. To validate this explanation further, in the right panel we increase the spread of the training points for each agent, so that the agents' training data overlap more. As expected, the standard errors are now much smaller—because there are multiple agents with training data in most regions, the procedure is more stable.

### D.2 Evaluation on Real Data

*Datasets.* The specifications of the datasets used for our experiments in Section 4.2 can be found in Table 2. The datasets have been downloaded from libsvm [Fan, 2011] and are used without any additional preprocessing, apart from the boston dataset that is normalized for neural network training. The boston housing price data [Dua and Graff, 2017] and the E2006 datasets [Kogan et al., 2009] have continuous outcome variables. Abalone [Dua and Graff, 2017] has integer values outcomes from $1 - 29$, YearPrediction [Dua and Graff, 2017] has integer valued year numbers from $1922 - 2011$, and the labels of cpusmall correspond to integers from $0 - 99$.

### Details on Benchmark Study in Table 1:

*Baselines.* We compare DeGroot to three baselines: M-avg, CV-static, CV-dynamic. The three methods describe different approaches to determine the weight $w_j$ of each model on the final

Table 2: Regression datasets used for evaluation, downloaded from [Fan, 2011].

|  | # samples | # featues |
|---|---|---|
| Boston | 506 | 13 |
| E2006 | 16087 | 3308 |
| Abalone | 4077 | 8 |
| cpusmall | 8192 | 12 |
| YearPrediction | 463715 | 90 |

prediction $p(x') = \sum_j w_j f_j(x')$. (M-avg) corresponds to the most natural baseline of equally weighted averaging with $w_j = \frac{1}{K}$ for all $j \in [K]$. This baseline that is optimal if models are unbiased and have equal variance. (CV-static) and (CV-adaptive) are two methods that have access to additional hold-out data to evaluate the predictive accuracy $\text{MSE}_j$ of each model $j \in [K]$ and determine the weights $w_j$. As the name says (CV-static) uses static weights, and (CV-adaptive) uses an adaptive weighting scheme. Similar to DeGroot both methods use weights that are inversely proportional to the MSE of the models:

$$w_j = \frac{1/\text{MSE}_j}{\sum_j 1/\text{MSE}_j}$$

In CV-static $\text{MSE}_j$ is evaluated on the hold-out data, and in CV-adaptive $\text{MSE}_j$ is evaluated on a subset of the hold-out data, composing of the $N$ closest points to $x'$. We use the same distance measure, and number of neighbors $N$ as we use to perform cross-validation on the individual agents in DeGroot. Our goal is to minimize potential confounding for a most meaningful comparative investigation.

*Evaluation.* In all experiments the data is first randomly partitioned into train,test and validation data. The validation data is completely ignored by DeGroot and M-avg and only used for CV-static and CV-adaptive. The size of the hold-out data is equal to the size of one data partition. We use $n_{test} = \max(0.15n, 500)$ test samples. For every random split all methods are evaluated on the same set of local models. We tune the hyparparameter of each model using a rough hyperparameter search upfront, but our main goal is not to get optimal accuracy with each individual model, but rather to investigate how the different aggregation procedures can deal with (resonable) given models. Hyperparameters are reported in Table 1.

*Models.* For training of the models we use the built in model classes for scikit-learn [Pedregosa et al., 2011]. Ridge and Lasso are imported from `sklearn.linear_model` and the hyperparameter $\lambda$ corresponds to the regularizer strength. For the DTR we use the `DecisionTreeRegressor` from `sklearn.tree` where the hyperparameter max_depth corresponds to the maximum depth of the regression tree. For the neural network (NN) we use the `MLPRegressor` class from `sklearn.neural_network`, it uses relu activation by default, and we choose two hidden layers with 8 neurons each, and $\alpha$ corresponds to the regularization strength. All other parameters are set to their default values. Whenever a NN model has not converged within the given number of iterations (500 for E2006 and 200 for boston) the model is nevertheless included in the ensemble since the purpose of this study is the evaluate how the different schemes deal with given models that might be heterogeneous in terms of quality.

*Confidence Intervals.* The reported confidence intervals are over the randomness in the data splitting and partitioning, they show one standard deviation from the mean.

**Additional Evaluations:**

*Heterogeneity.* We perform additional evaluations with the different approaches of achieving heterogeneity described in Section 4.2. In Figure 9 and Figure 10 we show results for ridge regression on the abalone, boston and cpusmall dataset. The gain of DeGroot over M-avg consistently grows with the degree of heterogeneity.

*Scalability.* In Figure 11 we investigate the scaling of the DeGroot method on two different datasets. We show results for 1 up to 128 agents. The training data (after test set hold-out) is partitioned across the agents by partially sorting the label with $p = 0.5$. This means with increasing number of agents $K$ the partitions get smaller. We choose the number of neighbors in DeGroot as $N =$

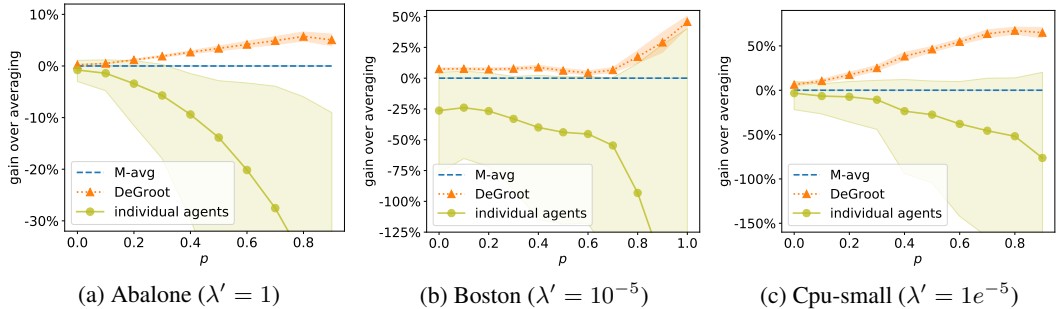

(a) Abalone ($\lambda' = 1$)  (b) Boston ($\lambda' = 10^{-5}$)  (c) Cpu-small ($\lambda' = 1e^{-5}$)

Figure 9: *(Label heterogeneity)*. Relative gain of DeGroot over M-avg. Same setting as Figure 3a for ridge regression on different datasets. We depict the performance of the individual models in the ensemble using the shaded green area, the upper edge corresponds to the best and the lower edge to the worst performing model, the green line indicating the average performance across the models.

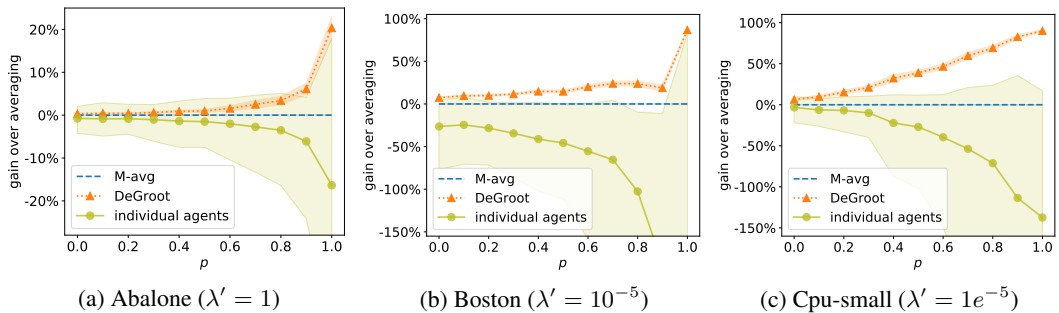

(a) Abalone ($\lambda' = 1$)  (b) Boston ($\lambda' = 10^{-5}$)  (c) Cpu-small ($\lambda' = 1e^{-5}$)

Figure 10: *(Feature heterogeneity)*. Relative gain of DeGroot over M-avg. Same setting as Figure 3b for ridge regression on different datasets. We partially sort the training data along feature 8 (Abalone) feature 9 (Boston) and feature 11 (cpusmall). The performance of the individual models in the ensemble is depicted using the shaded green area, the upper edge corresponds to the best and the lower edge to the worst model, the green line indicating the average performance across the models.

$\max(2, 0.01 * n_{\text{local}})$, where $n_{\text{local}}$ denotes the size of a local partition. We find that DeGroot scales robustly with the number of agents in the ensemble and outperforms averaging by a large margin. Remarkably, the adaptive weighting of DeGroot is still effective for 128 agents on the abalone datasets, where each agent only has access to 30 samples.

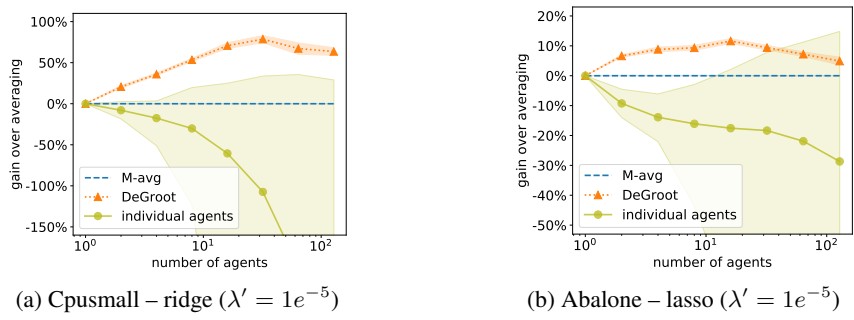

(a) Cpusmall – ridge ($\lambda' = 1e^{-5}$)  (b) Abalone – lasso ($\lambda' = 1e^{-5}$)

Figure 11: *(Scaling behavior of DeGroot)*. Relative gain of DeGroot aggregation over model averaging (M-avg) for different number of agents. Experiments performed for two different configurations.

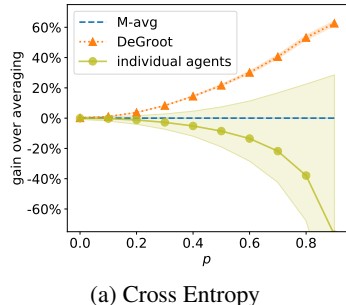

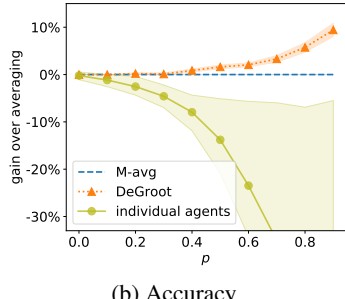

(a) Cross Entropy

(b) Accuracy

Figure 12: *(Binary Classification).* Relative gain of DeGroot aggregation over model averaging (M-avg). The value of $p$ is a proxi for the level of heterogeneity in the data partitioning.

### D.3 Classification example

In this section we outline how the DeGroot aggregation procedure could be used in combination with classification tasks. The reason why we have focused on regression in the main body of the paper is because it gives us a better handle on optimality which is appealing from an analysis point of view. However there is nothing that prevents us from using the DeGroot consensus procedure for also solving classification tasks. It remains to define an appropriate score for determining mutual trust.

For simplicity, let us consider a binary classification task with $y \in \{0, 1\}$ and let the pre-trained models be $L_2$-regularized logistic regression models. The individual predictions $f_i(x')$ correspond to the predicted probability $\widehat{p}_i(x') = p_i(Y = 1 | X = x')$ of model $i$. Then, to evaluate the trustworthiness of other models in Step 5 of Algorithm 1 we use the cross entropy loss instead of mean squared error. This is a natural score to use for classification, but there might also be other sensible choices. Formally, this means we replace (3) by:

$$\text{Loss}_{i,j}(x') = \frac{1}{|\mathcal{D}_i(x')|} \sum_{(x,y) \in \mathcal{D}_i(x')} \text{I}[y = 1] \log(\widehat{p}_j(x)) + \text{I}[y = 0] \log(1 - \widehat{p}_j(x))$$

and we compute the trust scores as

$$\tau_{ij} = \frac{1/\text{Loss}_{ij}}{\sum_{j \in [K]} 1/\text{Loss}_{ij}}. \tag{7}$$

Then, following the procedure outlined in Algorithm 1 we apply DeGroot to iteratively pool the individual predictions, using the trust scores as weights.

To evaluate the performance of DeGroot aggregation for this classification task empirically, we reproduce the heterogeneity experiment from Section 4.2 on the Phishing datasets for binary classification downloaded form the LIBSVM website [Fan, 2011]. Results are depicted in Figure 12. Here sorting the labels along the $y$-axis implies that the label imbalance in agents' partitions increases with $p$. We plot the relative gain in cross entropy as well as predictive accuracy of DeGroot over model averaging and compare it to individual agents as in Section 4.2. These results suggest that the consistent gains we have observed for regression also translate to classification.