# OpenReview forum: "Test-time Collective Prediction"
_NeurIPS.cc/2021/Conference — NeurIPS 2021 Poster_

### Official Review · Reviewer_gYkb · 2021-07-16

**Rating:** 7
**Confidence:** 3

**Summary:**

This paper considers collective prediction at test time, and proposes a novel decentralized mechanism by leveraging each agent's pretrained model and data without sharing them with other agents, inspired by human consensus in social science. In particular, for each test example, the proposed approach computes trust scores between two agents and aggregate the scores for a final collective prediction via DeGroot aggregation. In theory, this paper proves that the proposed DeGroot consensus mechanism is asymptotically optimal under some conditions. The efficacy of the proposed approach is empirically demonstrated by showing that the proposed scheme is better than an un-weighted consensus baseline.

The major contributions include the following:
(1) this paper introduces the concept of "mutual trust scores among agents" in social science (which is similar to PageRank algorithm in data science) for collective prediction at test time,
(2) the proposed approach is advantageous over existing methods since it does not require agents to share data or model parameters, which are beneficial in terms of communication cost and privacy perspective,
(3) the proposed approach is advantageous over existing methods since it does not require additional hold-out validation set to aggregate predictions of each agent,
(4) the proposed approach is advantageous over existing methods since it does not have any restriction on training procedure and a trained model of each agent,
(5) this paper theoretically proves the optimality of the proposed scheme (i.e., DeGroot consensus mechanism) is optimal under some conditions, and
(6) this paper empirically demonstrates that the proposed approach is superior than the baseline approach (which does not weight the predictions of agents).

**Limitations And Societal Impact:**

Yes.


**Main Review:**

**Originality:**
This paper introduces, for the first time, the known DeGroot consensus model in social science to address collective prediction at test time in the "decentralized" environment. PageRank can be considered as one application of the DeGroot consensus model, but it is not specifically used for collective predictions at test time.

I think the key claim is that the proposed approach has its "decentralization" properties, e.g., each agent does not need to share its data and model to conclude the final, collective prediction via predictions from multiple agents. But, one concern in computing mutual scores in Eq (2) is that each agent actually needs to see other agents' local data (assuming models are not shared); in particular, Eq (2) means that the i-th agent computes the mutual scores of other agents by evaluating the j-th agent prediction over the i-th agent data, which implies the i-th agent data needs to be transferred to the j-th agent for the prediction.
- Considering this, the paper's claim that each agent's data is not shared (e.g., line 35) could be overstated, and its unclear that the proposed approach addresses learning/inference issues in decentralized settings (since actually data sharing happens).
- Assuming each agent need to transmit 1% of its data (as mentioned in line 611), i.e., |D_i(x')| = 0.01|D_i|. in Eq (2), each agent can obtain most of other agent's covariate data about 100 inference iterations. This is equivalent to having one agent all data without labels across agents, then what's the benefit of the proposed approach compared to conventional semi-supervised learning approaches, i.e., one agent just trains a model using its labeled data and unlabeled data from other agents, or the ensemble of agent's models, each of which is trained using semi-supervised learning?


**Quality:**
This paper looks technically sound, and empirically well-justified on the importance of estimating "weight" of agent's predictions in aggregation.


**Clarity:**
This paper is clearly-written.
- line 609: is "1 − 10%" typo?

**Significance:**
I guess the DeGroot consensus idea in collective prediction could fertilize interesting ideas in other domains (e.g., classification or uncertainty estimation under distribution shift, where ensemble of models could be effective, but the models are combined with equal-weights).

**Time Spent Reviewing:**

5

---

> ### Author Response · Authors · 2021-08-10
> **Response to Reviewer gYkb**
>
> We thank the reviewer for the time and effort in reviewing our paper.
>
> We acknowledge that the claim in line 35 was overstated. We have rephrased it to be more explicit: Our algorithm does not require agents to share any labeled data. It is correct that agent $i$ needs to send its feature vector to agent $j$ to get agent $j$'th prediction for evaluating the mutual trust score $\tau_{ij}$.
>
> However, in many applications the labels contain the privacy-sensitive information and sharing the feature vector can be more acceptable from a privacy perspective. For example, in a healthcare setting the feature vector may be some set of demographic measurements, whereas the response may be the (much more sensitive) disease status. Moreover, we highlight that sharing of the unlabeled feature vectors is decentralized and as-needed; there is no central agent that collects all the data in advance. This is a far more limited form of data sharing than approaches that require centralized data pooling.
>
> It is an interesting question how our approach would compare to semi-supervised learning. The strength of our proposed approach is that it allows many agents with different local data to combine their knowledge to perform well for many different inputs. Instead, semi-supervised learning would struggle to achieve this for a high degree of heterogeneity across agents. Different datasets can live in different regions of the input space, the support of the data partitions can be non-overlapping and in this case unlabeled data does not help the agents to learn a better model.
>
> As a concrete example, consider a two-agent setting where Agent 1 has 1-dimensional features ($X$) lying only in the interval $[-2, -1]$ and Agent 2 has data only in the interval $[1, 2]$. If the response $Y$ is the absolute value of $X$, then Agent 1 with labeled data from $[-2, -1]$ and unlabeled data from $[1, 2]$ would be unable to make good predictions on input points falling in the interval $[1, 2]$. By contrast, the DeGroot consensus mechanism would result in good predictions on any input point in $[-2, -1]$ or $[1, 2]$, since the aggregated predictions are able to use all labels and upweight good predictions accordingly. In summary, to get optimal performance, we want to take advantage of all the labeled data available in the system -- this is exactly the task the DeGroot scheme is designed to solve.
>
> Nonetheless, as the reviewer pointed out, our proposed procedure does result in sharing of unlabeled feature vectors, which could be leveraged by the agents to improve their models in certain cases. We had not considered this possibility before, as we started from pre-trained local models and do not assume any form of retraining. But since our procedure is agnostic to the agent's modeling choices, we will mention this suggestion as a possible extension of the proposed method. Thank you for pointing this out!

---

> > ### Comment · Reviewer_gYkb · 2021-08-27
> > **Response**
> >
> > Thanks for the reply and the concrete example in semi-supervised learning. I think my major concerns can be addressed and highlighted via the final revision and additional discussion. So, I raise my score from 5 to 7 in the favor of the interesting DeGroot consensus model.

---

### Official Review · Reviewer_tLXB · 2021-07-17

**Rating:** 8
**Confidence:** 3

**Summary:**

This paper proposes an interesting approach based on DeGroot’s consensus model for combining predictions of multiple pre-trained models to collectively make a final prediction. The proposed method is useful in cases where data pooling, model sharing, or external validation is difficult. Experiments show that the proposed method is better than classical baselines of model averaging under heterogeneous conditions, without assuming access to additional labeled data for validation or learning a meta-model.


**Limitations And Societal Impact:**

Yes

**Main Review:**

Pros:
- The motivation of collective prediction from multiple independent models without model/weight sharing has a lot of practical relevance. Thus, this paper might be of interest to a broader community.
- The proposed method does not require additional validation data for learning aggregation to do collective predictions and scales well with the number of models in the ensemble.
- Sufficient and convincing experiments in the main paper and appendix demonstrating the usefulness of the proposed method for regression tasks. The authors also analyze the conditions when the proposed method may not be significantly better than simple ensembling approaches.

Limitations:
- Experiments are limited to the task of regression and involve only simple record-based datasets. The experimental section could be made stronger with more complex regression/classification tasks.
- Computing trust scores for each test example might affect the latency in comparison to other simpler baselines.

Questions:
- Is there any specific reason why classification tasks were not considered?




**Time Spent Reviewing:**

3.5

---

> ### Author Response · Authors · 2021-08-10
> **Response to Reviewer tLXB**
>
> We thank the reviewer for the comments and the effort in reviewing our paper.
>
> The reason why our manuscript focuses on regression problems is because in this context we have some handle on optimality: using the inverse MSE as the trust scores results in the optimal weighting scheme in some cases, as stated formally in Theorem 3.6.
> The algorithm proposed in this work does also apply to classification tasks. However, for classification, we do not have a corresponding result about optimality. There is not as natural of a score to use for determining the optimal set of weights -- although there are certainly sensible choices that would be reasonable in practice, such as the local classification accuracy or cross entropy loss. In any case, we will include this discussion of classification in the paper. Should the reviewer find it valuable, we can also include an empirical demonstration of the method with a classification task in the manuscript.

---

> > ### Comment · Reviewer_tLXB · 2021-08-28
> > **After Rebuttal**
> >
> > Thank you for the response. My ratings remain unchanged.
> > I believe that including the results for classification tasks would help in a better understanding of this method.

---

### Official Review · Reviewer_9CMX · 2021-07-18

**Rating:** 7
**Confidence:** 3

**Summary:**

This paper addresses test-time collective prediction. As far as I know, this is an underexplored topic, yet it is interesting. The paper proposes a decentralized consensus algorithm for multi-agent model aggregation. The paper includes both a nice theoretical analysis and empirical validation. Although I did not check the proofs, the theoretical analysis seems sound.

**Limitations And Societal Impact:**

Yes, included social impact.

**Main Review:**

Overall, I tend to favor accepting this paper, so I am tentatively voting with a 7. However, I have some concerns below that I would like the authors to clarify.

KEY CONCERN:
I would like the authors to clarify one aspect of the proposed algorithm. It seems like the trust model relies on the agent's querying each other on a validation subset of their private data. How does this not violate the premise that the agents are not willing to share data: "[the mechanism] does not require any data sharing, communication of model parameters, or labeled validation data."?

With respect to the above, I might recommend that the authors modify the protocol such that the collaborative agents use cryptographic tools like secure 2-party communication or FHE in order for agent i and agent j to agree on MSE_ij without agent i learning what agent j's model predicts on any particular data point in D_i(x') and without agent j learning what any particular data point in D_i(x') is. By only agreeing on MSE_ij, agent i and j are sharing the minimum amount of information necessary for the protocol to work.

The authors should either (1) clarify why this type of data sharing still is acceptable given their own premise,  (2) modify the protocol as described above (or any other way that achieves the objective laid out in their premise), or (3) clarify how I am misunderstanding the protocol if there is no data sharing.

**Time Spent Reviewing:**

2

---

> ### Author Response · Authors · 2021-08-10
> **Response to Reviewer 9CMX**
>
> We thank the reviewer for the detailed review and we will comment on the main concerns in the following:
>
> We acknowledge that the statement "the algorithm does not require any data sharing" (line 35) was too strong. We have rephrased it to be more explicit: Our algorithm does not require agents to share any _labeled_ data. As the reviewer pointed out, agent $i$ needs to send its feature vectors $D_i(x')$ to agent $j$ to get agent $j$'th predictions for evaluating MSE_ij.
>
> In many applications the labels contain the privacy-sensitive information and sharing the feature vector is acceptable from a privacy perspective. For example, in a healthcare setting the feature vector may be some set of demographic measurements, whereas the response may be the (much more sensitive) disease status. Moreover, we highlight the sharing of the unlabeled feature vectors is decentralized and as-needed; there is no central agent that collects all the data in advance. This is a far more limited form of data sharing than approaches that require centralized data pooling.
>
> We like the reviewer's point that it would be interesting to augment the proposed algorithm with cryptographic tools to further protect the information contained in the feature vector $X$ or the model queries. A different option would be to use techniques from differential-privacy and add an appropriate level of noise to the information that is shared. There is no reason why the proposed procedure could not be carried out with these extensions. The only downside if we use privatized feature vectors and queries is that mutual trust scores will be less accurate, but they likely still contain useful signals. We will add a discussion of these potential approaches of how to share minimal information and protect information in $X$ to the next version of the manuscript.
>
> At the same time, our focus in this work lies on the conceptual idea of introducing the DeGroot consensus-finding model in a machine learning setting. It has appealing properties from a privacy perspective (such as the fact that no raw labels need to be shared, and agents remain in control of how much information they incrementally give away throughout the algorithm), but we are aware that we do not provide any rigorous privacy guarantees. We intentionally focused on evaluating the value of this decentralized aggregation scheme in isolation, and we would like to defer a thorough development of rigorous privacy guarantees to a future investigation.

---

> > ### Comment · Reviewer_9CMX · 2021-08-25
> > **Reviewer response**
> >
> > Thanks for the reply. Agreed with the possible extensions using crypto or differentially privacy as an interesting future direction. I maintain my score.

---

### Official Review · Reviewer_rCjs · 2021-07-19

**Rating:** 7
**Confidence:** 4

**Summary:**

The authors present a decentralized algorithm for test-time model combination (collective prediction) wherein each party has an private model and dataset, and only has query access to other party's models. The algorithm is a novel application of DeGroot's consensus model which models how several agents iteratively influence each other's predictions based on mutual trust. The authors use the mean squared error on a local set of points as a proxy for mutual trust: each party queries each other party's models using points neighboring the test instance from its private dataset and measures the MSE on this set. The authors also show that the algorithm easily lends itself to a decentralized jackknife algorithm.

The authors prove several intuitive properties of their algorithm and prove that, asymptotically, it recovers an optimal inverse-MSE weighting. Empirically, they show that the method outperforms a naive equally-weighted average of the classifier predictions on synthetic and real data. Through synthetic experiments, the authors show that the proposed algorithm is particularly beneficial when there is significant heterogeneity in the private data or learners because of the adaptively computed weighting (based on MSE in a neighborhood around the test instance).

**Limitations And Societal Impact:**

The author should mention or address the limitation regarding agents leaking private data by providing query access mentioned above.

I do not foresee any negative societal impact from this work.

**Main Review:**

This is a well-written, well-executed paper. The authors present a novel algorithm and characterize it both theoretically and through experiments. The experiments highlight where the model is expected to work well and where it reduces to existing baselines. The authors also provide an excellent survey or related work.

My main concern is the motivation for this learning setup: the key motivation for decentralized collective prediction is to keep allow parties to keep their data and models private. However, as parties use their own data to query other models, they are in affect sharing their dataset piece by piece with others. Is it possible to share data between parties in a differentially private way?

Minor typos:

Lines 116-119: recommend using \(x\) instead of \(x'\).

**Time Spent Reviewing:**

3

---

> ### Author Response · Authors · 2021-08-10
> **Response to Reviewer rCjs**
>
> We thank the reviewer for the encouraging comments and for pointing out the inexact claim regarding data sharing.
>
> We acknowledge that the statement 'the algorithm does not require any data sharing' (line 35) was too strong. We have rephrased it to be more explicit that only unlabeled data needs to be shared with other agents to perform the local cross-validation procedure.
> We want to highlight that the sharing of the unlabeled feature vectors is decentralized and as-needed; there is no central agent that collects all the data in advance. This is a far more limited form of data sharing than approaches that require centralized data pooling.
>
> Taking this one step further, we agree with the reviewer that it would be interesting to augment the proposed algorithm with differentially-private sharing of the feature vectors to further protect the information contained in $X$. There is no reason why the proposed procedure could not be carried out with privatized feature vectors, with the only downside being that trust scores are less accurate -- but likely still useful. By adding the right level of noise, one could obtain explicit differential-privacy guarantees which would be very attractive. This is an interesting extension and we will make sure to include a discussion of this point in the light of the limitations of the current method in the next version of the manuscript.

---

### Decision · Program_Chairs · 2021-09-27

**Decision:**

Accept (Poster)

**Comment:**

Strengths:
- Novel approach and application of the DeGroots consensus model
- Thorough theoretical analysis
- Effective empirical demonstration
- As stated by one reviewer: “well-written, well-executed”

Weaknesses (mild):
- Motivation for decentralized collection prediction could be improved
- The way in which privacy and data sharing properties of the approach are described

Summary:

Reviewers were unanimous in their view that this is a strong and complete submission. This was evident both in the original reviews as well as in correspondences with the authors. Authors are encouraged to discuss in the final version issues raised by the reviewers, including (i) how privacy and data sharing are considered, (ii) experimenting with classification (in addition to regression), and (iii) possible future relations to crypto and differential privacy.